# Biotransformed Metabolites of the Hop Prenylflavanone Isoxanthohumol

**DOI:** 10.3390/molecules24030394

**Published:** 2019-01-22

**Authors:** Hyun Jung Kim, Soon-Ho Yim, Fubo Han, Bok Yun Kang, Hyun Jin Choi, Da-Woon Jung, Darren R. Williams, Kirk R. Gustafson, Edward J. Kennelly, Ik-Soo Lee

**Affiliations:** 1College of Pharmacy and Natural Medicine Research Institute, Mokpo National University, Muan, Jeonnam 58554, Korea; hyunkim@mokpo.ac.kr; 2Department of Pharmaceutical Engineering, Dongshin University, Naju, Jeonnam 58245, Korea; virshyim@dsu.ac.kr; 3College of Pharmacy, Chonnam National University, Gwangju 61186, Korea; hanfubo0306@gmail.com (F.H.); bykang@chonnam.ac.kr (B.Y.K.); 4College of Pharmacy and Institute of Pharmaceutical Sciences, CHA University, Seongnam, Gyeonggi-do 13488, Korea; hjchoi3@cha.ac.kr; 5New Drug Targets Laboratory, School of Life Sciences, Gwangju Institute of Science and Technology, Gwangju 61005, Korea; jung@gist.ac.kr (D.-W.J.); darren@gist.ac.kr (D.R.W.); 6Molecular Targets Program, Center for Cancer Research, National Cancer Institute, Frederick, MD 21702-1201, USA; gustafki@mail.nih.gov; 7Department of Biological Sciences, Lehman College, City University of New York, Bronx, NY 10468, USA; edward.kennelly@lehman.cuny.edu

**Keywords:** microbial transformation, hop prenylflavanone, isoxanthohumol

## Abstract

A metabolic conversion study on microbes is known as one of the most useful tools to predict the xenobiotic metabolism of organic compounds in mammalian systems. The microbial biotransformation of isoxanthohumol (**1**), a major hop prenylflavanone in beer, has resulted in the production of three diastereomeric pairs of oxygenated metabolites (**2**–**7**). The microbial metabolites of **1** were formed by epoxidation or hydroxylation of the prenyl group, and HPLC, NMR, and CD analyses revealed that all of the products were diastereomeric pairs composed of (2*S*)- and (2*R*)- isomers. The structures of these metabolic compounds were elucidated to be (2*S*,2″*S*)- and (2*R*,2″*S*)-4′-hydroxy-5-methoxy-7,8-(2,2-dimethyl-3-hydroxy-2,3-dihydro-4*H*-pyrano)-flavanones (**2** and **3**), (2*S*)- and (2*R*)-7,4′-dihydroxy-5-methoxy-8-(2,3-dihydroxy-3-methylbutyl)-flavanones (**4** and **5**) which were new oxygenated derivatives, along with (2*R*)- and (2*S*)-4′-hydroxy-5-methoxy-2″-(1-hydroxy-1-methylethyl)dihydrofuro[2,3-*h*]flavanones (**6** and **7**) on the basis of spectroscopic data. These results could contribute to understanding the metabolic fates of the major beer prenylflavanone isoxanthohumol that occur in mammalian system.

## 1. Introduction

Isoxanthohumol (**1**) (5-methoxy-8-prenylnaringenin, C_21_H_22_O_5_) is a well-known prenylated flavanone which occurs together with the prenylated chalcone xanthohumol in the female inflorescences (cones) of *Humulus lupulus* L. (hops) (Cannabaceae), which are added during the beer brewing process [1,2]. This flavanone has been specifically regarded as a beer prenylflavanone, since it is the main isomeric product of xanthohumol cyclization formed during hop processing and brewing. Hops naturally contain only minor quantities of isoxanthohumol compared with those of the most abundant hop chalcone xanthohumol, whereas beer contains much higher levels of isoxanthohumol than xanthohumol [3,4]. Biological and pharmacological properties of isoxanthohumol (**1**) have been less characterized than those of xanthohumol, but **1** has shown moderate estrogenic activity [5], antiproliferative and anticancer activities [6,7,8], cancer chemoprevention properties [9], and modifying effects in ontogenetic steroidogenesis [10].

Despite its importance in beer, only a few metabolism studies have been carried out with isoxanthohumol (**1**) to identify its metabolic fate in humans. Nikolic and colleagues investigated the oxidative metabolism of **1** using human liver microsomes in vitro and described several metabolites on the basis of liquid chromatography-tandem mass spectrometry [11]. Modification on one of the two terminal methyl groups of the prenyl moiety into the *cis*- and *trans*-hydroxymethyl analogues, respectively, followed by further oxidation to the *cis*- and *trans*-aldehydes, and double bond migration and subsequent hydroxylation to give an *exo*-methylene with an allylic alcohol was observed. Derivatives formed by hydroxylation or oxidation on the A- or B-ring, and *O*-demethylation were also reported [11]. A recent metabolism study of **1** showed that it was transformed by microorganisms into derivatives with a dihydrofuran ring or a methoxyglucosyl group [12]. In addition, some metabolic studies have focused on the conversion or activation of isoxanthohumol into 8-prenylnaringenin, a potent phytoestrogen [13,14,15].

Microbial biotransformation studies are regarded as one of the most useful tools to mimic and predict the xenobiotic metabolism of compounds in mammalian systems. Clark and Hufford have systematically summarized and reviewed the potential for the microorganisms as tools in the study of drug metabolisms with a number of specific examples that demonstrated the similarity in microbial and mammalian metabolism of xenobiotics [16,17]. They noted that microbial systems could offer a reliable, reproducible alternative to small laboratory animals for preliminary drug metabolism studies to identify the structural modifications by enzymatic reactions. General techniques and methods utilized in microbial metabolism studies clearly offer the practical advantages of convenient and inexpensive maintenance, production of metabolites in high yields and considerable amounts, and curtail the sacrifice of animals in biomedical research [16,17,18]. However, despite all the strengths and interesting parallels enumerated, microbial biotransformation could not ever completely replace the validity of xenobiotic metabolism studies with animals as well as liver microsomes or perfused livers. This model is a recently accounted practical tool with high potential for the creation of molecular diversity far beyond the metabolic changes observed in mammals [18].

We previously reported that microbial biotransformation of hop prenylflavonoids, including xanthohumol and 8-prenylnaringenin, produced several glucosylated, acyl-glucosylated, and cyclized metabolites [19,20], while biotransformation of **1** provided a diastereomeric pair of metabolites that resulted from 7-*O*-glucosylation on the A-ring via microbial Phase II conjugation reaction [21]. In our ongoing metabolism study of hop prenylflavonoids, a preparative-scale microbial transformation of isoxanthohumol (**1**) by the fungi, *Rhizopus oryzae* KCTC 6399 and *Fusarium oxysporum* f.sp. *lini* KCTC 16325, afforded three pairs of oxygenated metabolites (**2**–**7**) (Figure 1). The production of these microbially biotransformed metabolites of **1** and their structure elucidation are reported herein.

## 2. Results and Discussion

### 2.1. Preparation and Microbial Biotransformation of Isoxanthohumol

Preparation of the substrate isoxanthohumol (**1**) was achieved by both chemical cyclization of xanthohumol in aqueous alkali solution [4] and enzymatic cyclization using a microbial transformation method [19]. The isoxanthohumol produced by these methods was a racemic mixture of (2*S*)- and (2*R*)-flavanones, which was confirmed by spectroscopic data analysis showing no absorption in the CD spectrum and no optical rotation.

A total of forty-one microbial cultures were screened for their ability to metabolize **1** and two fungi, *Rhizopus oryzae* KCTC 6399 and *Fusarium oxysporum* f.sp. *lini* KCTC 16325, were selected for further scale-up fermentation studies. For each microbe, separate substrate and culture control studies were carried out under the same fermentation conditions, which showed that the metabolites were produced as a result of enzymatic activity by the fungi, and not as a consequence of chemical or non-metabolic conversion. The ability to biotransform **1** was confirmed on the basis of reversed-phase (C_18_) TLC analyses. The *R*_f_ values of the three new diasteromeric mixtures (**2**,**3**: *R*_f_ 0.32, **4**,**5**: 0.42, and **6**,**7**: 0.28) were significantly larger than that of **1** (*R*_f_ 0.15), which indicated that the fungi produced metabolites with higher polarity.

### 2.2. Structure Elucidation of Isoxanthohumol Metabolites

Metabolites **2** and **3** of isoxanthohumol were produced by the fungus *R. oryzae* KCTC 6399 and obtained as a pale yellow amorphous powder. HPLC analyses of the mixture of **2** and **3** revealed that each metabolite had the same ultraviolet (UV) spectral data at 190–400 nm and a minor difference of retention time (*t*_R_) (**2**: 21.56 min and **3**: 22.45 min). Their ^1^H and ^13^C-NMR spectra also exhibited nearly identical chemical shift and coupling constant values, which suggested that they consisted of a pair of diastereomers. HRESIMS of metabolites **2** and **3** exhibited [M + H]^+^ peaks at *m*/*z* 371.1504 and *m*/*z* 371.1482 (calcd for C_21_H_23_O_6_, 371.1495) respectively, which established their molecular formula as C_21_H_22_O_6_ and indicated that they were mono-oxygenated metabolites of **1**. The UV spectrum of both compounds displayed typical absorptions for a flavanone, with a maximum absorption peak at ~287 nm and an inflection at 318 nm, which were similar to those observed for **1**. However, their ^1^H-NMR spectra showed major differences in the isoprenyl group, which showed two methyl signals shifted upfield (H-4″ and H-5″) at *δ*_H_ 1.31, 1.28 (3H, s) and 1.33, 1.34 (3H, s) as well as a new oxymethine signal (H-2″) at *δ*_H_ 3.76 (brt, *J* = 6.0 Hz) and 3.73 (1H, dd, *J* = 7.3, 5.8 Hz) in **2** and **3**, respectively. These observations suggested that the two sp^2^ carbons of the isoprenyl group in **1** had been metabolized to oxygenated sp^3^ carbons. The ^13^C-NMR data of both compounds showed resonances for an sp^3^ oxymethine at *δ*_C_ 69.6 and 69.9 (C-2″) and an oxygenated sp^3^ quaternary carbon at *δ*_C_ 79.9 (C-3″), and the loss of the olefin signals at *δ*_C_ 124.0 and 131.9 that were observed in **1**. An oxygenated aromatic carbon at *δ*_C_ 161.9 (C-7) in metabolites **2** and **3** was shifted upfield relative to the corresponding carbon in **1** (*δ*_C_ 164.0). These results indicated that a dihydropyran ring had been formed by oxidation and subsequent cyclization of the isoprenyl moiety, possibly going through an epoxide intermediate. Two-dimensional NMR data, including HSQC and HMBC experiments, supported the presence of a dihydropyran ring. The *gem*-dimethyl protons H_3_-4″ and H_3_-5″ correlated with C-2″ and C-3″, and the H_2_-1″ methylene protons correlated with C-8 and C-3″. Based on these spectroscopic analyses, the planar structure of diastereomeric metabolites **2** and **3** was assigned as 4′-hydroxy-5-methoxy-7,8-(2,2-dimethyl-3-hydroxy-2,3-dihydro-4*H*-pyrano)-flavanone. The absolute configurations of the two asymmetric carbons, C-2 and C-2″, were established by circular dichroism [22,23], and the Mosher’s ester method [24,25], respectively. Metabolite **2** displayed positive and negative Cotton effects at 331 (n → π* transition) and 288 nm (π → π* transition), respectively, which corresponded to a 2*S* configuration. In contrast, metabolite **3** showed a negative Cotton effect at 334 nm and a positive one at 288 nm which established a 2*R* configuration (See Appendix A). The absolute configuration at C-2″ was determined by the modified Mosher’s method. Metabolite **3** was converted into the (*S*)- and (*R*)-methoxytrifluoromethylphenylacetic acid (MTPA) esters, **3a** and **3b**, by treatment with (*R*)- and (*S*)-MTPA chloride, respectively. The Δ*δ* values (*δ*_H_ = *δ_S_* − *δ_R_*, ppm) calculated for the two esters (Figure 2) indicated that C-2″ had an *S* configuration. From these results, unambiguous structures of **2** and **3** were assigned as (2*S*,2″*S*)- and (2*R*,2″*S*)-4′-hydroxy-5-methoxy-7,8-(2,2-dimethyl-3-hydroxy-2,3-dihydro-4*H*-pyrano)-flavanones, respectively.

Metabolites **4** and **5** of isoxanthohumol were also obtained by the enzymatic activity of *R. oryzae* KCTC 6399 and isolated as a pale yellow amorphous powder. As observed with metabolites **2** and **3**, HPLC analyses (*t*_R_, **4**: 11.65 min and **5**: 12.56 min) and spectroscopic data including UV, infrared (IR), ^1^H- and ^13^C-NMR spectra indicated that two diastereomeric isomers were produced. Metabolites **4** and **5** exhibited HRESIMS [M + H]^+^ peaks at *m*/*z* 389.1612 and *m*/*z* 389.1600 (calcd for C_21_H_25_O_7_, 389.1600) respectively, which suggested a molecular formula of C_21_H_24_O_7_ corresponding to dihydroxylated derivatives of **1**. The UV spectra of **4** and **5** displayed characteristic absorptions of a flavanone moiety at 225, 285 and ~310 nm, which were similar to their substrate **1**. However, the ^1^H-NMR spectra of **4** and **5** exhibited major differences in the isoprenyl group resonances. They showed two pairs of methyl signals shifted upfield (H-4″ and H-5″) at *δ*_H_ 1.12, 1.15 (3H, s) and 1.14, 1.16 (3H, s), as well as oxymethine signals (C-2″) at *δ*_H_ 3.52 (dd, *J* = 10.0, 2.3 Hz) and 3.55 (dd, *J* = 10.0, 2.0 Hz), suggesting formation of two hydroxylated sp^3^ carbons from the olefinic group of **1**. The ^13^C-NMR spectra of both compounds showed the presence of a hydroxymethine signal (C-2″) at *δ*_C_ 80.2 in **4** and *δ*_C_ 80.1 in **5**, and an oxygenated quaternary carbon at *δ*_C_ 74.1 (C-3″) in both. In addition, ^13^C-NMR signals of a *gem*-dimethyl group at *δ*_C_ 25.0 and 26.0, were consistent with their substitution on an oxygenated carbon. HMBC correlations from H_3_-4″ and H_3_-5″ to C-2″ and C-3″, as well as from H_2_-1″ to C-8 and C-2″, and H-2″ to C-8 helped to establish the structure of diastereomers **4** and **5** as 7,4′-dihydroxy-5-methoxy-8-(2,3-dihydroxy-3-methylbutyl)-flavanone. The absolute configuration of the asymmetric center at C-2 was assigned by CD studies [22,23]. Metabolite **4** displayed positive and negative Cotton effects at 331 (n → π* transition) and 289 nm (π → π* transition), respectively, corresponding to a 2*S* configuration, while C-2 in metabolite **5** was established to have a 2*R* configuration from the opposite Cotton effects at 331 (negative) and 288 nm (positive). Therefore, the structures of metabolites **4** and **5** were assigned as (2*S*)- and (2*R*)-7,4′-dihydroxy-5-methoxy-8-(2,3-dihydroxy-3-methylbutyl)-flavanones, respectively. Regarding the determination of absolute configuration at C-2″, an asymmetric center of oxygenated *C*-prenyl side chain in aryl prenyl derivatives, X-ray crystallography has been considered as the most powerful approach as shown in a case of a prenylated coumarin, meranzin hydrate [26]. Adequate single crystals of **4** and **5** suitable for X-ray diffraction studies, however, were not available due to the limitations of their physicochemical properties and limited quantities. Chiroptical tools including electronic circular dichroism (ECD) were not effectively applicable on account of the strong Cotton effect curves obtained from (2*S*)- and (2*R*)-flavanones [23,27] (See Appendix A). In an effort to determine the absolute configuration at C-2″, Mosher’s ester derivatives were prepared, however the results were ambiguous. The *bis*-MTPA esters of **4** and **5** provided Δ*δ* values (*δ*_H_ = *δ_S_* − *δ_R_*) that had a non-uniform distribution of positive and negative signs, which meant that Mosher’s analysis was not valid in this case [24,25]. Further, chemical synthesis may be necessary to determine absolute configuration of the chiral carbon C-2″ at the prenyl moiety.

Metabolites **6** and **7** were also obtained as a pair of diastereomers by microbial transformation of **1** using *F. oxysporum* f.sp. *lini*, which was confirmed by HPLC (*t*_R_, **6**: 18.63 min and **3**: 19.42 min) and NMR experiments. HRESIMS of **6** and **7** showed [M + Na]^+^ peaks at *m*/*z* 393.1307 and *m*/*z* 393.1316 (calcd for C_21_H_23_O_6_Na, 393.1314), respectively, which established their molecular formula as C_21_H_22_O_6_ which was isomeric with the mono-oxygenated metabolites **2** and **3**. However, several significant differences were observed in the ^1^H- and ^13^C-NMR spectra of **6** and **7**. Highly downfield shifted oxymethine proton (*δ*_H_ 4.73, brt, *J* = 8.8 Hz and 4.74, dd, *J* = 9.3, 8.0 Hz) and carbon (*δ*_C_ 92.5) resonances of C-2″ indicated that a dihydrofuran ring was formed from the isoprenyl group substituted on the A-ring. The NMR data of **6** and **7** were in good agreement with those of (*2S*)-4′-hydroxy-5-methoxy-7,8-[2-(1-hydroxy-1-methylethyl)-2,3-dihydrofurano]flavanone, a metabolite of xanthohumol by *Pichia membranifaciens* (ATCC 2254) [28], and (*2R*)-4′-hydroxy-5-methoxy-7,8-[2-(1-hydroxy-1-methylethyl)-2,3-dihydrofurano]flavanone, a metabolite of isoxanthohumol by *F. equiseti* (AM15) [12]. The structures of compounds **6** and **7** were assigned as 4′-hydroxy-5-methoxy-2″-(1-hydroxy-1-methylethyl)dihydrofuro[2,3-*h*]flavanones, and the absolute configurations of the asymmetric center at C-2 were identified by CD [22,23]. Metabolite **6** showed negative and positive Cotton effects at 325 and 289 nm, respectively, corresponding to 2*R*, while C-2 in metabolite **7** was established to be 2*S* from the opposite sign Cotton effects at 324 (positive) and 290 nm (negative). The configuration at C-2″ in compounds **6** and **7** could not be assigned from the NMR or CD measurements.

## 3. Materials and Methods

### 3.1. General Experimental Procedures

Optical rotations were recorded with a JASCO DIP 1000 digital polarimeter (JASCO, Tokyo, Japan). UV spectra were recorded on a JASCO V-530 spectrophotometer (JASCO, Tokyo, Japan), and CD spectra were recorded on a JASCO J-810 spectrometer (JASCO, Tokyo, Japan). IR spectra were obtained on a JASCO FT/IR 300-E spectrometer (JASCO, Tokyo, Japan). ^1^H-, ^13^C-, HSQC, and HMBC NMR experiments were recorded using a Varian Unity INOVA 500 spectrometer (Agilent Technologies, Inc., Santa Clara, CA, USA). HRESIMS were determined on Waters Synapt HDMS LC/MS mass spectrometer (Waters Corp., Milford, MA, USA). TLC was carried out on Merck silica gel F_254_-precoated glass and RP-18 F_254S_ plates (Merck, Darmstadt, Germany). Medium pressure liquid chromatography (MPLC) was performed using Lobar^TM^ C_18_ column (10 × 240 mm, 40–63 μm, Merck, Darmstadt, Germany) and silica gel (40–63 μm, Merck). HPLC was performed on a Hewlett-Packard Agilent 1100 Series (Agilent Technologies, Inc., Santa Clara, CA, USA) HPLC System composed of a degasser, a binary mixing pump, a column oven and a DAD detector using Waters SunFire™ (Waters Corp., Milford, MA, USA) (4.6 × 150 mm, 5 μm) and SunFire™ Prep C_18_ column (10 × 150 mm, 5 μm) with acetonitrile (solvent A) and water containing 0.1% formic acid (solvent B).

### 3.2. Chemicals and Ingredients

Isoxanthohumol was prepared by chemical cyclization in aqueous NaOH solution at 0 °C as described by Stevens et al. [4], and also by a microbial transformation method using the fungus *R. oryzae* KCTC 6946 as previously reported by Kim and Lee [19]. Isoxanthohumol prepared by both methods was extracted with EtOAc and then purified by chromatographic methods including silica gel and reversed-phase C_18_ MPLC. The spectroscopic data of isoxanthohumol (**1**) were in good agreement with data in the literature [1] and its structure was also confirmed by 2D NMR experiments. Optical rotation and CD measurements revealed that the substrate isoxanthohumol was a racemic mixture of (*2S*)- and (*2R*)-isoxanthohumol. Ingredients for media including D-glucose, peptone, malt extract, yeast extract, and potato dextrose medium were purchased from Becton, Dickinson and Co. (Sparks, MD, USA), and sucrose was purchased from Sigma-Aldrich Co. (St Louis, MO, USA).

### 3.3. Microorganisms and Fermentation

Forty-one microbial strains were obtained from the Korean Collection for Type Cultures (KCTC) and cultured for preliminary screening. Microorganisms and culture broth composition were described in the previous literature in detail [20,21].

### 3.4. Biotransformation Screening Procedure

All of the microbial cultures were grown according to the two-stage procedure [16,17]. In the screening studies, the actively-growing microbial cultures were inoculated in 100 mL flasks containing 20 mL of media, and incubated with gentle agitation (200 rpm) at 25 °C in a temperature-controlled shaking incubator. Isoxanthohumol (**1**) (2 mg/0.1 mL in EtOH) was added to each flask 24 h after inoculation, and further fermented under the same condition for 3 d. Sampling and TLC monitoring were generally carried out on RP-18 TLC_254S_ with 60% MeOH at 24 h intervals. Two control studies were performed for identification of metabolites produced by enzymatic transformation. Substrate controls consisted of **1** and each sterile medium incubated without microorganisms. Culture controls consisted of fermentation cultures in which the microorganisms were grown without addition of **1**.

### 3.5. Biotransformation of Isoxanthohumol *(**1**)* by R. oryzae KCTC 6399

Preparative-scale fermentations were carried out under the same condition with two 1 L flasks each containing 250 mL of medium and 20 mg of isoxanthohumol (**1**) for 10 d. The cultures were extracted with EtOAc two times and the organic layers were combined and concentrated at reduced pressure. The EtOAc extract (730 mg) was subjected to silica gel (70–230 mesh, Merck) column chromatography with a CHCl_3_–MeOH (9:1) to give three fractions. Fraction 1 (108 mg) was chromatographed by RP-MPLC (Lobar^TM^, 10 × 240 mm) using 55% MeOH isocratic solvent system to give a mixture of isoxanthohumol metabolites **2** and **3** (13.2 mg, 15.8% yield). An aliquot of compounds (7.8 mg) was further chromatographed by HPLC with a gradient solvent system of 20% solvent A to 33% solvent A for 25 min to afford two isomers **2** (1.6 mg, *t*_R_ 21.56 min) and **3** (2.2 mg, *t*_R_ 22.45 min). Fraction 2 (203 mg) was also chromatographed by RP-MPLC (Lobar^TM^, 10 × 240 mm) using aqueous MeOH solvent (50 → 55%) to give a mixture of metabolites **4** and **5** (7.7 mg, 8.9% yield). A portion of mixture (4.3 mg) was further purified by HPLC with a gradient solvent system of 20% A to 35% A for 25 min to afford two isomers **4** (1.4 mg, *t*_R_ 11.65 min) and **5** (1.6 mg, *t*_R_ 12.56 min).

*(2S*,*2*″*S)*-*4*′-*Hydroxy*-*5*-*methoxy*-*7*,*8*-*(2*,*2*-*dimethyl*-*3*-*hydroxy*-*2*,*3*-*dihydro*-*4H*-*pyrano)*-*flavanone* (**2**): pale yellow amorphous powder, [α]_D_ +37.0° (*c* 0.2, MeOH); UV λ_max_ (MeOH) (log ε) 224 (4.35) 285 (4.18), 319 (3.65) nm; CD (MeOH) λ_ext_ (Δε): 288 (−13.5), 311 (0.0), 331 (+4.7); IR (KBr) ν_max_: 3421, 1658, 1608, 1579, 1519, 1485, 1338, 1206, 1133, 1106, 835 cm^−1^; ^1^H-NMR (CD_3_OD, 500 MHz) δ 7.33 (2H, d, *J* = 8.3 Hz, H-2′, 6′), 6.82 (2H, d, *J* = 8.3 Hz, H-3′, 5′), 6.09 (1H, s, H-6), 5.37 (1H, brd, *J* = 12.5 Hz, H-2), 3.80 (3H, s, 5-OC*H*_3_), 3.76 (1H, brt, *J* = 6.0 Hz, H-2″), 3.02 (1H, dd, *J* = 16.5, 13.0 Hz, H-3a), 2.82 (1H, dd, *J* = 17.0, 5.0 Hz, H-1″a), 2.70 (1H, dd, *J* = 16.5, 3.0 Hz, H-3b), 2.55 (1H, dd, *J* = 17.0, 6.5 Hz, H-1″b), 1.33 (3H, s, H-5″), 1.31 (3H, s, H-4″); ^13^C-NMR (CD_3_OD, 125 MHz) δ 192.7 (C-4), 164.1 (C-8a), 162.0 (C-5), 161.9 (C-7), 159.1 (C-4′), 131.4 (C-1′), 129.0 (C-2′,6′), 116.5 (C-3′,5′), 106.4 (C-4a), 101.9 (C-8), 94.9 (C-6), 80.4 (C-2), 79.9 (C-3″), 69.6 (C-2″), 56.3 (5-O*C*H_3_), 46.2 (C-3), 26.7 (C-1″), 25.7 (C-5″), 22.0 (C-4″); ESIMS *m*/*z* 371 [M + H]^+^; HRESIMS *m*/*z* 371.1504 [M + H]^+^ (calcd for C_21_H_23_O_6_, 371.1495).

*(2R*,*2*″*S)*-*4*′-*Hydroxy*-*5*-*methoxy*-*7*,*8*-*(2*,*2*-*dimethyl*-*3*-*hydroxy*-*2*,*3*-*dihydro*-*4H*-*pyrano)*-*flavanone* (**3**): pale yellow amorphous powder, [α]_D_ +66.4° (*c* 0.2, MeOH); UV λ_max_ (MeOH) (log ε) 224 (4.47), 288 (4.30), 318 (3.77) nm; CD (MeOH) λ_ext_ (Δε): 288 (+16.4), 313 (0.0), 334 (−5.4); IR (KBr) ν_max_: 3421, 1658, 1608, 1578, 1519, 1485, 1337, 1206, 1133, 1106, 835 cm^−1^; ^1^H-NMR (CD_3_OD, 500 MHz) δ 7.32 (2H, d, *J* = 8.5 Hz, H-2′, 6′), 6.82 (2H, d, *J* = 8.5 Hz, H-3′, 5′), 6.08 (1H, s, H-6), 5.36 (1H, dd, *J* = 13.0, 2.5 Hz, H-2), 3.79 (3H, s, 5-OC*H*_3_), 3.73 (1H, dd, *J* = 7.3, 5.8 Hz, H-2″), 3.00 (1H, dd, *J* = 16.5, 13.0 Hz, H-3a), 2.83 (1H, dd, *J* = 17.0, 5.5 Hz, H-1″a), 2.69 (1H, dd, *J* = 16.5, 3.0 Hz, H-3b), 2.50 (1H, dd, *J* = 17.0, 7.0 Hz, H-1″b), 1.34 (3H, s, H-5″), 1.28 (3H, s, H-4″); ^13^C-NMR (CD_3_OD, 125 MHz) δ 192.7 (C-4), 163.9 (C-8a), 162.0 (C-5), 161.9 (C-7), 159.1 (C-4′), 131.4 (C-1′), 129.0 (C-2″,6′), 116.5 (C-3′,5′), 106.4 (C-4a), 102.2 (C-8), 94.9 (C-6), 80.3 (C-2), 79.9 (C-3″), 69.9 (C-2″), 56.3 (5-O*C*H_3_), 46.2 (C-3), 26.7 (C-1″), 25.9 (C-5″), 21.3 (C-4″); ESIMS *m*/*z* 371 [M + H]^+^; HRESIMS *m*/*z* 371.1482 [M + H]^+^ (calcd for C_21_H_23_O_6_, 371.1495).

*(2S)*-*7*,*4*′-*Dihydroxy*-*5*-*methoxy*-*8*-*(2*,*3*-*dihydroxy*-*3*-*methylbutyl)*-*flavanone* (**4**): pale yellow amorphous powder, [α]_D_ −31.7° (*c* 0.2, MeOH); UV λ_max_ (MeOH) (log ε): 225 (4.42), 285 (4.17), 310 (3.73) nm; CD (MeOH) λ_ext_ (Δε): 289 (−9.1), 313 (0.0), 331 (+4.0); IR (KBr) ν_max_: 3424, 1600, 1515, 1463, 1351, 1280, 1150, 1099, 828 cm^−1^; ^1^H-NMR (CD_3_OD, 500 MHz) δ 7.34 (2H, d, *J* = 8.3 Hz, H-2′, 6′), 6.81 (2H, d, *J* = 8.3 Hz, H-3′, 5′), 6.16 (1H, s, H-6), 5.33 (1H, brd, *J* = 13.5 Hz, H-2), 3.81 (3H, s, 5-OC*H*_3_), 3.52 (1H, dd, *J* = 10.0, 2.3 Hz, H-2″), 3.00 (1H, dd, *J* = 16.8, 13.5 Hz, H-3a), 2.93 (1H, brd, *J* = 14.0 Hz, H-1″a), 2.69 (1H, brd, *J* = 16.8 Hz, H-3b), 2.62 (1H, dd, *J* = 14.0, 10.0 Hz, H-1″b), 1.14 (3H, s, H-5″), 1.12 (3H, s, H-4″); ^13^C-NMR (CD_3_OD, 125 MHz) δ 192.9 (C-4), 165.5 (C-7), 164.2 (C-8a), 162.5 (C-5), 159.0 (C-4′), 131.5 (C-1′), 129.0 (C-2′,6′), 116.4 (C-3′,5′), 108.5 (C-8), 105.9 (C-4a), 94.6 (C-6), 80.4 (C-2), 80.2 (C-2″), 74.1 (C-3″), 56.1 (5-O*C*H_3_), 46.3 (C-3), 26.6 (C-1″), 26.0 (C-4″), 25.0 (C-5″); ESIMS *m*/*z* 389 [M + H]^+^; HRESIMS *m*/*z* 389.1612 [M + H]^+^ (calcd for C_21_H_25_O_7_, 389.1600).

*(2R)*-*7*,*4*′-*Dihydroxy*-*5*-*methoxy*-*8*-*(2*,*3*-*dihydroxy*-*3*-*methylbutyl)*-*flavanones* (**5**): pale yellow amorphous powder, [α]_D_ −53.0° (*c* 0.2, MeOH); UV λ_max_ (MeOH) (log ε): 225 (4.30), 285 (4.08), 318 (3.65) nm; CD (MeOH) λ_ext_ (Δε): 288 (+10.8), 311 (0.0), 331 (−3.8); IR (KBr) ν_max_: 3422, 2975, 1599, 1514, 1463, 1349, 1281, 1210, 1150, 1101, 829 cm^−1^; ^1^H-NMR (CD_3_OD, 500 MHz) δ 7.34 (2H, d, *J* = 8.5 Hz, H-2′, 6′), 6.81 (2H, d, *J* = 8.5 Hz, H-3′, 5′), 6.17 (1H, s, H-6), 5.34 (1H, dd, *J* = 13.0, 2.5 Hz, H-2), 3.82 (3H, s, 5-OC*H*_3_), 3.55 (1H, dd, *J* = 10.0, 2.0 Hz, H-2″), 2.98 (1H, dd, *J* = 16.5, 13.0 Hz, H-3a), 2.95 (1H, dd, *J* = 14.0, 2.5 Hz, H-1″a), 2.70 (1H, dd, *J* = 16.5, 3.0 Hz, H-3b), 2.61(1H, dd, *J* = 14.0, 10.0 Hz, H-1″b), 1.16 (3H, s, H-5″), 1.15 (3H, s, H-4″); ^13^C-NMR (CD_3_OD, 125 MHz) δ 192.9 (C-4), 165.5 (C-7), 164.0 (C-8a), 162.6 (C-5), 159.0 (C-4′), 131.6 (C-1′), 128.9 (C-2′,6′), 116.4 (C-3′,5′), 108.6 (C-8), 105.8 (C-4a), 94.7 (C-6), 80.5 (C-2), 80.1 (C-2″), 74.1 (C-3″), 56.1 (5-O*C*H_3_), 46.4 (C-3), 26.7 (C-1″), 26.0 (C-4″), 25.0 (C-5″); ESIMS *m*/*z* 389 [M + H]^+^; HRESIMS *m*/*z* 389.1600 [M + H]^+^ (calcd for C_21_H_25_O_7_, 389.1600).

### 3.6. Biotransformation of ***1*** by F. oxysporum f.sp. lini KCTC 16325

Scale-up fermentations were carried out under the same condition with two 1 L Erlenmeyer flasks each containing 250 mL medium and 25 mg isoxanthohumol (**1**) for 5 d. Production of a metabolite was monitored by reversed-phase C_18_ TLC (MeOH 70%). The red-colored cultures were extracted with EtOAc two times and the organic layers were combined and concentrated in vacuo. The EtOAc extract (280 mg) was subjected to silica gel (70–230 mesh, Merck) column chromatography with *n*-hexane-EtOAc (2:1) mixture to give three fractions. Fraction 2 (89 mg) containing a metabolite was chromatographed with MPLC (Lobar^TM^, 10 × 240 mm) using MeOH 45% isocratic solvent system to afford isoxanthohumol as a mixture of metabolites **6** and **7** (18 mg, 34.3% yield). An aliquot of compound mixture (3.4 mg) was further chromatographed by HPLC with a gradient solvent system of 20% A to 35% A for 25 min to afford two isomers **6** (1.1 mg, *t*_R_ 18.63 min) and **7** (1.1 mg, *t*_R_ 19.42 min).

### 3.7. Determination of Absolute Configuration by Modified Mosher’s Method

Compound **3** (1.0 mg in 0.2 mL pyridine) was treated with 15 μL (20.3 mg) of (*R*)-(−)-α-methoxy-α-(trifluoromethyl)phenylacetyl chloride (MTPA-chloride), and stirred overnight at room temperature under nitrogen gas. The reaction mixture was evaporated in vacuo, and chromatographed by HPLC using a SunFire™ Prep C_18_ (10 × 150 mm, Waters) with a gradient solvent MeCN–H_2_O (70:30 → 90:10) at 3 mL/min for 20 min, afforded the (*S*)-Mosher ester derivative **3a** (*t*_R_ 17.30, 0.9 mg). The derivative of (*R*)-Mosher ester **3b** (*t*_R_ 17.49, 1.0 mg) was prepared by reaction of **3** (1.0 mg) with 15 μL of (*S*)-(−)-MTPA-chloride reagent under the same condition as described above.

**3a**: ^1^H-NMR (CDCl_3_, 500 MHz) δ 6.03 (1H, s, H-6), 5.44 (1H, dd, *J* = 12.6, 3.0 Hz, H-2), 5.16 (1H, dd, *J* = 6.0, 5.4 Hz, H-2″), 3.86 (3H, s, 5-OC*H_3_*), 2.99 (1H, dd, *J* = 17.4, 5.4 Hz, H-1″), 2.93 (1H, dd, *J* = 16.2, 12.6 Hz, H-3), 2.83 (1H, dd, *J* = 16.2, 3.0 Hz, H-3), 2.65 (1H, dd, *J* = 17.4, 6.0 Hz, H-1″), 1.36 (3H, s, H-5″), 1.32 (3H, s, H-4″). **3b**: ^1^H-NMR (CDCl_3_, 500 MHz) δ 6.06 (1H, s, H-6), 5.46 (1H, dd, *J* = 12.6, 3.0 Hz, H-2), 5.16 (1H, dd, *J* = 6.0, 5.4 Hz, H-2″), 3.87 (3H, s, 5-OC*H_3_*), 3.00 (1H, dd, *J* = 17.4, 5.4 Hz, H-1″), 2.95 (1H, dd, *J* = 16.2, 12.6 Hz, H-3), 2.87 (1H, *J* = 16.2, 3.0 Hz, H-3), 2.80 (1H, dd, *J* = 17.4, 5.4 Hz, H-1″), 1.31 (3H, s, H-5″), 1.27 (3H, s, H-4″).

## 4. Conclusions

Microbial transformation studies of isoxanthohumol (**1**) resulted in the production of three oxygenated pairs of metabolites (**2**–**7**). These include two novel metabolite pairs possessing a dihydropyran (**2**,**3**) and a dihydroxymethylbutane (**4**,**5**) substituted flavanone core structure. It was widely reported that cyclization at the prenyl side chain resulted in forming a five-membered furan or a six-membered pyran heterocycle attached to the A-ring in prenylated flavanones [29,30]. The formation of dihydroxymethylbutyl group has frequently occurred by dihydroxylation on the double bond of the prenyl group substituted to the A ring in biotransformation of prenylated flavanones [29]. These transformation and derivatization have been commonly encountered in natural products [31], and the presence of cyclized and dihydroxylated derivatives of the prenyl substituent has been revealed in the bark of the Amazonian tree *Brosimum acutifolium*, a rich source of 8-prenylated flavonoids [32].

Unlike the microbial metabolites **2**–**7**, the regioselectively mono-hydroxylated prenyl side chain metabolites of isoxanthohumol, namely *cis*- and *trans*-prenyl alcohols, were previously identified as the most abundant oxidation metabolites of isoxanthohumol during in vitro metabolism studies using human liver microsomes, which was catalyzed by hepatic cytochrome P450 enzymes [11,33]. However, knowledge of the in vitro microbial metabolic conversions of isoxanthohumol may contribute to the detection and identification of the metabolic products of isoxanthohumol that occur in mammalian systems. Conjugation reactions including sulfation and glucuronidation are involved in the major metabolic pathways of polyphenols in mammals, which metabolize polyphenols into very hydrophilic conjugates [34,35]. It was previously proved that microbial aryl sulfotransferase was capable of producing Phase II metabolites of flavonoids, rather close to the mammalian enzyme [35]. Considering that isoxanthohumol is a significant component of beer and that beer is consumed by a large number of people world-wide, these findings could have potential health and nutritional implications.

## Figures and Tables

**Figure 1 molecules-24-00394-f001:**
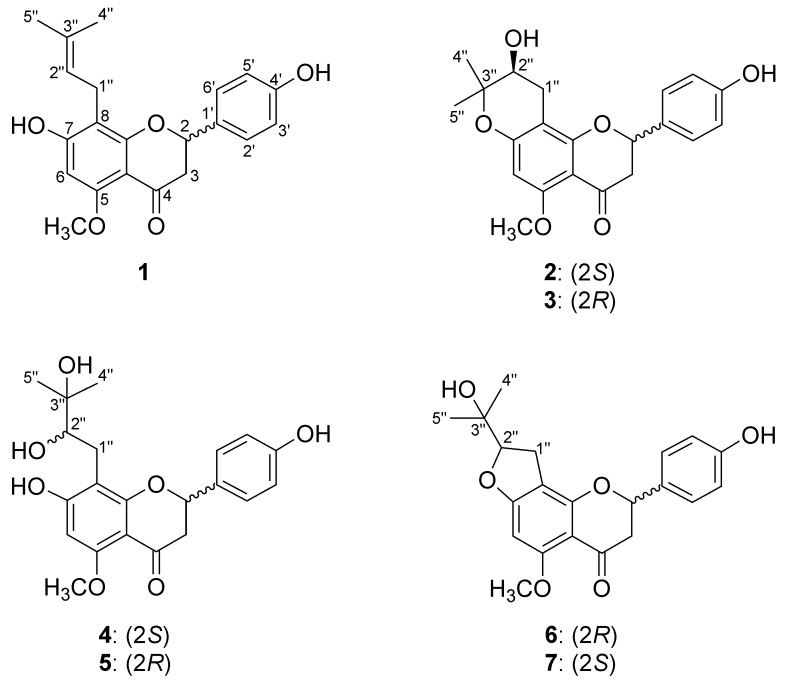
The structures of isoxanthohumol (**1**) and its metabolites (**2**–**7**).

**Figure 2 molecules-24-00394-f002:**
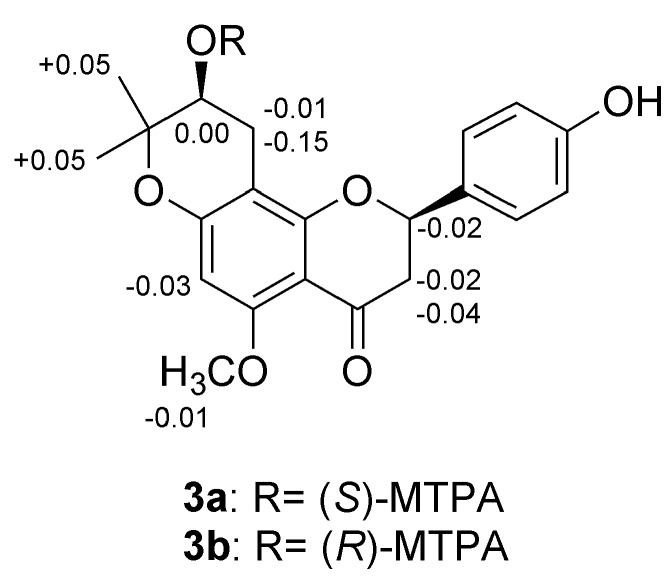
The Δδ*_S_*_−*R*_ values (ppm) from Mosher ester derivatives of **3**.

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
