# Peer review of "Biotransformed Metabolites of the Hop Prenylflavanone Isoxanthohumol"

_molecules, 2019, doi:10.3390/molecules24030394_

Round 1

Reviewer 1 Report

This manuscript treats metabolism of isoxanthohumol by two fungal species, and structures of six products were characterized. The manuscript was written well, although the following points should be considered by the authors.

1) Although the authors mentioned importance of metabolism in mammalian systems, they used fungal species, instead of intestinal bacteria. The authors should add the reason in the introduction.

2) Hydroxylation and cyclization of the prenyl group of flavonoids in plants have been reported by various groups. The authors should mention that and compare the results.

3) Position numbers used in compounds 2, 3, 6, and 7 were different from those in compounds 1, 4, and 5. For better understanding of their structural relations, they should be used in the same system.

Author Response

Point 1: Although the authors mentioned importance of metabolism in mammalian system, they used fungal species, instead of intestinal bacteria. The authors should add the reason in the introduction.

Response 1: The following paragraph was added to the Introduction:

Microbial biotransformation studies are regarded as one of the most useful tools to mimic and predict the xenobiotic metabolism of compounds in mammalian systems. Clark and Hufford have systematically summarized and reviewed the potential for the microorganisms as tools in the study of drug metabolism with a number of specific examples that demonstrated the similarity in microbial and mammalian metabolism of xenobiotics [16,17]. They noted that microbial systems could offer a reliable, reproducible alternative to small laboratory animals for preliminary drug metabolism studies to identify the structural modifications by enzymatic reactions. General techniques and methods utilized in microbial metabolism studies clearly offer practical advantages of convenient and inexpensive maintenance, production of metabolites in high yields and considerable amounts, and curtail the sacrifice of animals in biomedical research [16-18]. Moreover, microbial biotransformation studies using natural drugs as substrates are used to produce biologically active compounds involving stereo-selective and regio-specific conversions, as compared with synthetic methods [18].

Point 2: Hydroxylation and cyclization of the prenyl group of flavonoids in plants have been reported by various groups. The authors should mention that and compare the results.

Response 2: The following text was added to the Conclusion:

It was widely reported that cyclization at the prenyl side chain resulted in forming a five-membered furan or a six-membered pyran heterocycle attached to the A ring in prenylated flavanones [27,28]. The formation of dihydroxymethylbutyl group has frequently occurred by dihydroxylation on the double bond of the prenyl group substituted to the A ring in biotransformation of prenylated flavanones [27]. These transformation and derivatization have been commonly encountered in natural products [29], and the presence of cyclized and dihydroxylated derivatives of the prenyl substituent has been revealed in the bark of the Amazonian tree Brosimum acutifolium, a rich source of 8-prenylated flavonoids [30].

Point 3: Position numbers used in compounds 2, 3, 6, and 7 were different from those in compounds 1, 4, and 5. For better understanding of their structural relations, they should be used in the same system.

Response 3: The position numbers for all of the compounds are now the same.

Reviewer 2 Report

Authors describe microbial biotransformation of isoxanthohumol and describe then a series of potential metabolites. The work is quite well performed, namely the spectral part, which is meticulously accomplished. There are, however, couple of points to be clarified:

Natural isoxanthohumol is a racemic mixture of 2-R and 2-S isomers (e.g. exactly 1:1)?? During oxidation experiments you created diastereomers separable without chiral aid - did you observe any diastereomeric enrichment?? In case of compounds 4 and 5 - did you determine absolute configuration at their 2' positions (why not??).

I have some concern about your extraction procedure - extraction with EtOAc is rather selective for more lipohilic compounds and therefore you could miss some more polar metabolites - extraction using SPE (e.g. Amberlite XAD-2 or -4 or RP C-18) would be definitely better choice. Can you comment on it, please.

You assume in your work that the metabolites obtained with fungal cultures could be similar or identical (??) with human (mammalian) metabolites, but this it pure speculation. This type of compounds (polyphenols) are typically metabolized by Phase II biotransformation, e.g by the conjugation (sulfation, glucuronidation, methylation etc.), oxidative biotransformation is much rarer - often accounting only for ca 5 - 10 % of all metabolome. Can you comment on this, please? What type of enzyme(s) can be responsible in fungi for the metabolic changes described.

Technical points:

(2S)-7,4'-dihydroxy-5-m....  must be written correctly (2S)-7,4'-Dihydroxy-5-m.. The word starts with Dihydroxy... prefixes are not valid here (see also other analogous titles). In the references, plase add to each reference DOI (as required by the Journal).

Author Response

Point 1: Natural isoxanthohumol is a racemic mixture of 2-R and 2-S isomers (e.g. exactly 1:1)?? During oxidation experiments you created diastereomers separable without chiral aid - did you observe any diastereomeric enrichment?? In case of compounds 4 and 5 - did you determine absolute configuration at their 2' positions (why not??).

Response 1: Natural isoxanthohumol is generally produced by cyclization of xanthohumol, a major hop prenylated chalcone, in brewing process, and it is a racemic mixture shown to be almost identical (each level of isomer, 1:1). Oxidative diastereomeric pairs were successfully separated using HPLC system as shown in supplementary materials Scheme S1.

The sentence below explains why the absolute configuration was not assigned, and it was already present in the text:

The bis-MTPA esters of 4 and 5 provided Δδ values (δH= δS-δR) that had a non-uniform distribution of positive and negative signs, which meant that Mosher’s analysis was not valid in this case [24,25].

Point 2: I have some concern about your extraction procedure - extraction with EtOAc is rather selective for more lipohilic compounds and therefore you could miss some more polar metabolites - extraction using SPE (e.g. Amberlite XAD-2 or -4 or RP C-18) would be definitely better choice. Can you comment on it, please.

Response 2: It is our experience that extraction with EtOAc provides both lipophilic and polar metabolites from culture broths, and the use of SPE adsorbents does not significantly alter the profile of polar metabolites that are obtained.

Point 3: You assume in your work that the metabolites obtained with fungal cultures could be similar or identical (??) with human (mammalian) metabolites, but this it pure speculation. This type of compounds (polyphenols) are typically metabolized by Phase II biotransformation, e.g by the conjugation (sulfation, glucuronidation, methylation etc.), oxidative biotransformation is much rarer - often accounting only for ca 5 - 10 % of all metabolome. Can you comment on this, please? What type of enzyme(s) can be responsible in fungi for the metabolic changes described.

Response 3: Microbial biotransformation studies are regarded as one of the most useful tools to mimic and predict the xenobiotic metabolism of compounds in mammalian systems. The potential for the microorganisms as tools in the study of drug metabolism has been systematically summarized and reviewed with a number of specific examples that demonstrated the similarity in microbial and mammalian metabolism of xenobiotics [16,17]. Hydroxylated metabolites of prenylated flavonoids have been widely reported as secondary metabolites in hop cones or as oxidative biotransformed Phase I metabolites from the hop flavonoids [28]. Furthermore, oxidative biotransformation rather than conjugation is most likely to occur in the beer brewing process, so these are the important biotransformation products that are consumed in the course of beer consumption.same.

Point 4: (2S)-7,4'-dihydroxy-5-m....  must be written correctly (2S)-7,4'-Dihydroxy-5-m.. The word starts with Dihydroxy... prefixes are not valid here (see also other analogous titles). In the references, plase add to each reference DOI (as required by the Journal).

Response 4: Chemical names of the metabolites were corrected, and DOI was added to each reference.

Round 2

Reviewer 2 Report

Most of the issues was corrected, however I am not happy with replies to Q. 3 and Q1. Concerning the metabolites of isoxanthohumol - you are mixing two different points - changes during brewing and metabolism in mammals (and microbial metabolites). Your reply unfortunately lacks critical view. Microbial metabolites may be similar or identical to the mammal ones, and useful namely in preparation of chiral compounds, but AGAIN, this is mere speculation. One of the valid method is incubation of compounds e.g. with liver microsomes or perfused liver. This type of references is totaly missing in your work. Take as an example this study

J Mass Spectrom. 2005 Mar;40(3):289-99.

Please, make a decent search in literature and pls. add some CRITICAL acount on the mammal and microbial metabolites.

Q1 - OK you cannot determine abs. configuration with Mosher's esters, I assume that ECD is also impossible to use (Cotton effect missing -???) but there are also other methods. E.g. comparison of ECD spectra with analogous compounds, crystallization and X-ray. Please, kindly CRITICALLY asses all possibilities, otherwise your work will lack quite important point (chirality of the microbial metabolites).

Author Response

Point 1: Microbial metabolites may be similar or identical to the mammal ones, and useful namely in preparation of chiral compounds, but AGAIN, this is mere speculation. One of the valid method is incubation of compounds e.g. with liver microsomes or perfused liver. This type of references is totaly missing in your work. Take as an example this study

J Mass Spectrom. 2005 Mar;40(3):289-99.

Please, make a decent search in literature and pls. add some CRITICAL acount on the mammal and microbial metabolites.

Response 1:

1) The following text was added to the Introduction (Ln. #70-74, in blue) w/ references:

However, despite all the strengths and interesting parallels enumerated, microbial biotransformation could not ever completely replace the validity of xenobiotic metabolism studies with animals as well as liver microsome or perfused liver. This model is recently accounted practical tools to be highly potential for the creation of molecular diversity far beyond the metabolic changes observed in mammals [18].

18.  Venisetty, R.K.; Cidii, V. Application of microbial biotransformation for the new drug discovery using   natural drugs as substrates. Curr. Pharm. Biotechnol. 2003, 4, 153-167, DOI: 10.2174/1389201033489847.

2) The following text was added to the Conclusion (Ln. #331-334, in blue) w/ references:

Unlike the microbial metabolites 2-7, the cis- and trans-prenyl alcohols were previously identified as the most abundant oxidation metabolites of isoxanthohumol during in vitro metabolism studies using human liver microsomes, which was catalyzed by hepatic cytochrome P450 enzymes [11,33].

11.  Nikolic, D.; Li, Y.; Chadwick, L.R.; Pauli, G.F.; van Breeman, R.B. Metabolism of xanthohumol and isoxanthohumol, prenylated flavonoids from hops (Humulus lupulus L.), by human liver microsomes. J. Mass Spectrom. 2005, 40, 289-299, DOI: 10.1002/jms.753.

33.  Guo, J.; Nikolic, D.; Chadwick, L.R.; Pauli, G.F.; van Breemen, R.B. Identification of human hepatic cytochrome P450 enzymes involved in the metabolism of 8-prenylnaringenin and isoxanthohumol from hops (Humulus lupulus L.). Drug Metab. Dispos. 2006, 34, 1152-1159, DOI: 10.1124/dmd.105.008250.

Point 2: Q1 - OK you cannot determine abs. configuration with Mosher's esters, I assume that ECD is also impossible to use (Cotton effect missing -???) but there are also other methods. E.g. comparison of ECD spectra with analogous compounds, crystallization and X-ray. Please, kindly CRITICALLY asses all possibilities, otherwise your work will lack quite important point (chirality of the microbial metabolites).

Response 2:

1) The following text was added (Ln. #161-168, in blue) w/ references:

Regarding the determination of absolute configuration at C-2", an asymmetric center of oxygenated C-prenyl side chain in aryl prenyl derivatives, X-ray crystallography has been considered as the most powerful approach as shown in a case of a prenylated coumarin, meranzin hydrate [26]. Adequate single crystals of 4 and 5 suitable for X-ray diffraction studies, however, were not available due to the limitations of their physicochemical properties and limited quantities. Chiroptical tools including electronic circular dichroism (ECD) were not effectively applicable on account of the strong Cotton effect curves obtained from (2S)- and (2R)-flavanones [23,27].

23.  Slade, D.; Ferreira, D.; Marais, J.P.J. Circular dichroism, a powerful tool for the assessment of absolute configuration of flavonoids. Phytochemistry 2005, 66, 2177-2215, DOI: 10.1016/j.phytochem.2005.02.002.

27.  Vázquez, J.T. Features of electronic circular dichroism and tips for its use in determining absolute configuration. Tetrahedron: Asymmetry 2017, 28, 1199-1211, DOI: 10.1016/j.tetasy.2017.09.015.

2) The following text was added (Ln. #171-172, in blue)

Further, chemical synthesis may be necessary to determine absolute configuration of the chiral carbon C-2" at the prenyl moiety.

Round 3

Reviewer 2 Report

Authors replied most of points sufficiently. I would like to remind the authors, however, that polyphenols are metabolized in mammals mostly by conjugation with sulfate or glucuronate (Phase II biotransformation; this has not been mentioned in the paper at all) and these metabolites can be obtained also by microbial biotransformations,  however, authors may add one explanatory sentence during typesetting or proof corection concerning conjugation reaction  (e.g. ChemCatChem 7, 3152-3162 (2015)). Anyway, I am quite happy with the present manuscript and I propose its acceptance.